# Dr. CLIP: CLIP-Driven Universal Framework for Zero-Shot Sketch Image Retrieval

## ABSTRACT

The field of Zero-Shot Sketch-Based Image Retrieval (ZS-SBIR) is currently undergoing a paradigm shift, transitioning from specialized models designed for individual tasks to more general retrieval models capable of managing various specialized scenarios. Inspired by the impressive generalization ability of the Contrastive Language-Image Pretraining (CLIP) model, we propose a CLIP-driven universal framework (Dr. CLIP), which leverages prompt learning to guide the synergy between CLIP and ZS-SBIR. Specifically, Dr. CLIP is a multi-branch network based on the CLIP image encoder and text encoder, which can perfectly cover four variants of ZS-SBIR tasks (inter-category, intra-category, cross-datasets, and generalization). Moreover, we decompose the synergy into classification learning, metric learning, and ranking learning, as well as introduce three key components to enhance learning effectiveness. *i)* a forgetting suppression idea is applied to prevent catastrophic forgetting and constrains the feature distribution of the new categories in classification learning. *ii)* a domain balanced loss is proposed to address sample imbalance and establish effective cross-domain correlations in metric learning. *iii)* a pair-relation strategy is introduced to capture relevance and ranking relationships between instances in ranking learning. Eventually, we reorganize and redivide three coarse-grained datasets and two fine-grained datasets to accommodate the training settings for four ZS-SBIR tasks. The comparison experiments confirmed our method surpassed the state-of-the-art (SOTA) methods by a significant margin (**1.95%~19.14%**, mAP), highlighting its generality and superiority. Source code link: https://github.com/xxxxxx.git.

## CCS CONCEPTS

• **Information systems** → **Image search**; • **Computing methodologies** → **Matching**.

## KEYWORDS

Sketch-based image retrieval, Zero-shot learning, Contrastive language-image pre-training (CLIP), Universal Framework

## 1 INTRODUCTION

Image retrieval is an important area in multimodal data retrieval [11, 14–16]. With the widespread use of handwriting input devices,

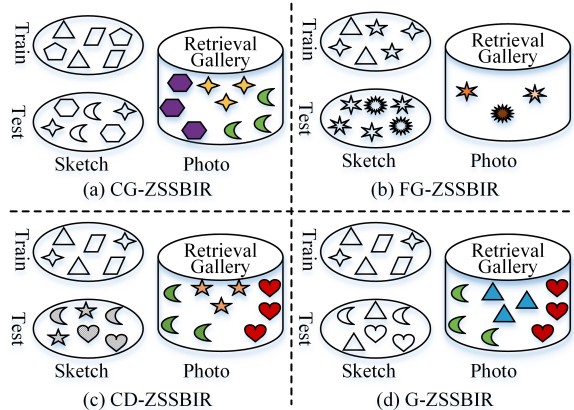

**Figure 1: ZS-SBIR research is primarily categorized into four scenarios: (a) zero-shot coarse-grained sketch-based image retrieval (ZS-CGSBIR), (b) zero-shot fine-grained sketch-based image retrieval (ZS-FGSBIR), (c) zero-shot cross-dataset sketch-based image retrieval, and (d) Generalized zero-shot sketch-based image retrieval (GZS-SBIR).**

sketch-based image retrieval (**SBIR**) have gained popularity. Existing SBIR methods typically require training for each category, which becomes impractical when dealing with a large number of categories [27]. Therefore, recent efforts have primarily focused on zero-shot sketch-based image retrieval (**ZS-SBIR**). This task trains on samples from seen categories and retrieves photo from unseen categories by using sketches as queries.

Existing ZS-SBIR scenarios can be classified into four types based on the gallery settings. As shown in Figure. 1(a), zero-shot coarse-grained sketch-based image retrieval (**ZS-CGSBIR**) aims to retrieve photos from a multi-category photo gallery where the categories of these photos are not seen during the model training [17, 19, 24, 25, 28, 29, 31, 38]. ZS-CGSBIR struggles to distinguish intra-category diversity, leading to the development of zero-shot fine-grained sketch-based image retrieval (**ZS-FGSBIR**) [2, 3, 20–22, 36]. As illustrated in Figure. 1(b), ZS-FGSBIR focuses on retrieving photos from gallery with fine-grained subcategories. In practical applications, we often encounter the need for image retrieval across different sources and styles of datasets. As depicted in Figure. 1(c), the model is first trained on one dataset and then tested on another dataset with non-overlapping categories. Zero-shot cross-dataset sketch-based image retrieval (**ZS-CDSBIR**) [17, 19, 21, 32] is designed to simulate this scenario more realistically. Generalized zero-shot sketch-based image retrieval (**GZS-SBIR**) [4–6, 10, 13, 40] extends the task to encompass both seen and unseen categories simultaneously. This means that during the query process, it is necessary not only to match sketches and photos from seen categories

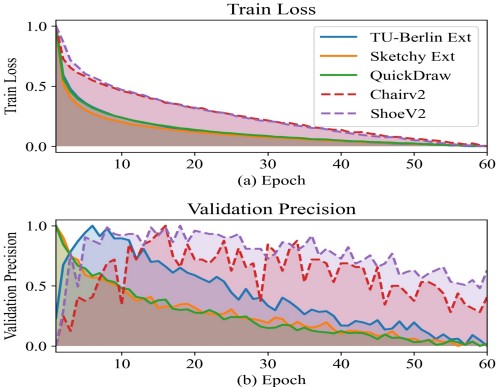

**Figure 2: The loss and precision curves of the CLIP-all method during training on five datasets.**

but also to accurately retrieve photos from unseen categories, as shown in Figure. 1(d). Despite the fact that most existing retrieval models are designed with a focus on specific tasks, there has been a noticeable lack of exploration in the development of a comprehensive framework that can effectively address these tasks collectively.

The CLIP model demonstrates remarkable cross-modal representation learning and the ability to construct a shared semantic space in multiple visual tasks [8, 25, 33, 39, 41]. It effectively reasons in zero-shot tasks. Therefore, we propose a CLIP-driven general framework for ZS-SBIR tasks targeting four specialized scenarios. Based on the aforementioned research, we identifiy two major challenges that constrain the synergy performance between CLIP and ZS-SBIR. i) During training, we notice that the loss consistently decreases across five datasets, but the validation accuracy shows a declining trend (Figure 2). We attribute this to disparity between CLIP's pre-training task and the specific requirements of ZS-SBIR hampers its adaptability, resulting in overfitting. ii) The imbalance between the number of samples in the photo domain and the sketch domain, both in terms of categories and overall quantity, can lead to a sample imbalance between the domains, which has been a lingering sample problem in SBIR.

To tackle the challenges mentioned above, we devise suitable objective functions and utilize a balanced cross-domain sampling strategy. We split the training objective of the Dr. CLIP model into classification learning, metric learning, and ranking learning, adopting the training approach of prompt learning [1]. The main contributions are as follows:

- We propose a novel CLIP-driven universal framework for multiple zero-shot sketch-image retrieval tasks, which aims to thoroughly explore the synergy between CLIP and ZS-SBIR. To the best of our knowledge, this is the first universal framework that covers four different ZS-SBIR subtasks.
- We introduce CLIP's existing knowledge distribution to constrain the distribution of new class semantic knowledge during training, and design a quadruplet loss to measure the similarity between sketch pairs and sketch-photo pairs, effectively mitigating catastrophic forgetting and domain sample imbalance issues.

- We utilize a fusion representation that combines textual features and image features to measure the semantic correlation between different categories. Multimodal feature representation enhances the model's ability to capture relevant and ordered relationships among images, resulting in more reasonable and logical search result rankings.
- We establish four ZS-SBIR experimental scenarios on three coarse-grained datasets and two fine-grained datasets. By comparing with state-of-the-art methods, the results demonstrate that our approach achieves the best retrieval accuracy in each task, showcasing its effectiveness and versatility.

## 2 RELATED WORK

### 2.1 Preliminaries Overview of CLIP

CLIP is a vision-language model developed by OpenAI [23], which is trained by performing contrastive learning on large-scale image and text data. It has the ability to understand both images and text simultaneously. The structure of the CLIP model is based on a shared vision-language encoder, denoted as F(·). Images are input to the image encoder as a sequence of K uniformly patch tokens, along with a class token and positional encoding, represented as $[E^I, CLS^I] \in R^{(k+1) \cdot d}$, and feature projection is performed on the class token to obtain the final visual representation $f_I = F_I(V^I)$. Sentences are input to the text encoder as embeddings of words, and they are encoded into fixed-length vector representations $f_T = F_T(V^T)$. Furthermore, CLIP exhibits potential in zero-shot learning as it learns shared semantic representations between images and text. By contrasting the description text of new categories with images, it enables zero-shot recognition of these new categories. This presents a novel solution for addressing ZS-SBIR tasks. Therefore, we aim to explore a CLIP-driven universal retrieval framework for multiple SBIR tasks in zero-shot scenarios.

### 2.2 ZS-CGSBIR

In recent years, several remarkable works have been proposed in ZS-CGSBIR. For instance, SAKE [19] advocates preserving inter-class relationships obtained from the original domain to achieve sketch-photo matching tasks. SketchGCN [38] and ACNet [24] both aim to reduce domain gaps and knowledge gaps for effective retrieval. TVT [29] and PSKD [31] introduce knowledge distillation, with the former aligning modalities through distillation and multimodal hypersphere learning, and the latter optimizing the student network using teacher signals extracted from a teacher network. Similar to TVT, TCN [30] also enhances the transferability of the network by aligning modality features effectively. Sketch3T [28] proposes a meta-learning framework that combines discriminative learning with auxiliary tasks, achieving high accuracy even without any additional training sketch-photo pairs. The methods most similar to our proposed approach are ZSE [17] and CLIP-all [25], both of which propose frameworks capable of simultaneously addressing coarse-grained and fine-grained ZS-SBIR tasks. The former is a convolutional model based on attention mechanisms, while the latter is based on the CLIP model. This work extends the ideas from [25] by optimizing the objective function through task decomposition, not only improving coarse-grained retrieval performance but also adapting to a wider range of ZS-SBIR tasks.

## 2.3 ZS-FGSBIR

This task requires the model to observe subtle differences among different subcategories, making it more challenging. In recent years, several outstanding works have been proposed in the field of FGS-BIR [2, 3, 20–22, 36]. For example, CC-DG [21] learns a universal fine-grained visual feature descriptor, enabling query sketches to provide appropriate auxiliary learning for the retrieval network. Triplet-RL [3] introduces a new reward scheme and proposes a reinforcement learning-based retrieval framework. MPA [18] introduces a discriminator-guided mechanism that treats generation and retrieval as two conjugate problems for joint learning. Progress has also been made in more challenging ZS-FGSBIR tasks. ZSE [17] is a more general retrieval framework that learns discriminative features at a finer granularity by establishing local patch correspondences and computing distance scores. CLIP-all [25] is also a more general retrieval framework that relies on the powerful category semantic knowledge of the CLIP model. Our work is inspired by these two general frameworks and modifies the original training objective of the CLIP-all model to enable it to align fine-grained features across domains.

## 2.4 ZS-CDSBIR

ZS-CDSBIR works [17, 19, 21, 32] have demonstrated higher usability and reliability in practical applications. For example, CC-DG [21] plays a crucial role in cross-dataset retrieval tasks by extracting instance-specific fine-grained features. DSN [32] effectively reduces the intra-class diversity in the sketch domain by mining relationships among additional augmented samples, smoothing the domain gap. The ZSE [17] model, known for its excellent ability to extract general features, is also suitable for sketch retrieval tasks across datasets. The setting of ZS-CDSBIR allows for a comprehensive evaluation of the model's generalization capabilities on completely different datasets. We have also validated our proposed method and obtained remarkably promising results.

## 2.5 GZS-SBIR

Compared to ZS-CDSBIR, the more generalized task of GZS-SBIR presents greater challenges and is more relevant to real-world applications. For example, SEM-PCYC [5], Doodle2Search [4], and OCEAN [40] all alleviate the domain gap by mapping sketches or photos into a shared semantic space. SEM-PCYC and OCEAN establish mapping relationships through adversarial training, while Doodle2Search introduces a novel strategy for mining inter-domain mutual information. STL [10] proposes a hybrid metric learning strategy to establish semantic-aware ranking attributes and calibrate the joint embedding space. There are also methods that differ from the above ideas. AMF [13] explores a novel knowledge discovery module to simulate new knowledge unseen during training, coupling the cross-domain distribution between photos and sketches in the visual space. StyleGuide [6] presents a detection method based on precomputed unseen class prototypes, where query sketches are compared only with photo data that is more likely to belong to unseen classes, thereby improving retrieval performance. Inspired by the above methods, we combine the ideas of constructing a shared semantic space and aligning cross-domain distributions to learn

new knowledge while retaining the original experience, thereby achieving effective retrieval.

## 3 THE PROPOSED METHOD

### 3.1 Zero Shot Setting

The dataset we utilized consists of sketches and photos $I = \{(S_i, P_j)|i = 0, \cdots, N^S. j = 0, \cdots, N^P\}$. The categories, denoted as $C$, are divided into two distinct sets $C = \{(C_j^{seen}, C_j^{unseen}) | j = 1, \cdots, N^C\}$: visible classes $C^{seen}$ and unseen classes $C^{unseen}$, with no overlap between them ($C^{seen} \cap C^{unseen} = \emptyset$). In the zero-shot scenario, the training set comprises sketches and photos of the visible classes, represented as $D_{train} = \{S^{seen}, P^{seen}\}$. Here, $S^{seen} = \{(s_i, y_i) | y_i \in C^{seen}\}$ represents the sketch of the visible classes, and $P^{seen}$ represents the corresponding photos. The test set comprises sketches and photos of the unseen classes, denoted as $D_{test} = \{S^{seen}, P^{seen}\}$. Similarly, $S^{unseen} = \{(s_i, y_i) | y_i \in C^{unseen}\}$ represents the sketch of the unseen classes, while $P^{unseen}$ represents the corresponding photos.

### 3.2 Overall Architecture

As shown in Figure. 3. To guide CLIP in learning the distributions of sketches and photos effectively, we introduce sketch and photo prompts in the input, represented as $V = \{(V^s, V^p) | V \in R^{NC \times d}\}$. Through backpropagation, the knowledge learned by CLIP is incorporated into the weights of the prompts. During the training process, while the parameters of layer normalization layers and prompts weight in the transformer encoder layers are trainable, and the remaining weights of the CLIP model are keep frozen. So, specific prompts inject the model to generate sketch features $f_S = F_S(S, V^s)$ and photo features $f_P = F_P(S, V^p)$.

To mitigate the risk of overfitting during the training, we also feed the inputs to the frozen original CLIP image encoder to obtain the raw features $(f'_S, f'_P)$, which are then used to supervise the training of both branches. Additionally, unlike traditional retrieval models with classification heads, Dr. CLIP differs in that we attempt to utilize the text encoder of CLIP to further leverage natural language features $f_T$ for obtaining category semantic information. This enables a richer semantics representation of the textual data and enhances the model's ability to understand and interpret the underlying semantics of the photos and sketches. We utilize the prompt template 'a photo of a [$class$]' to obtain sentences related to the labels, where [$class$] needs to be replaced with the real class name. Finally, we employ the acquired image and text features to undertake a series of learning tasks, including classification learning, metric learning, and ranking learning. These tasks collectively improve the model's ability to generalize and enhance its capability to represent discriminative features during the training process.

### 3.3 Forgetting Suppression Classification Loss

For classification learning, we utilize cross-entropy (CE) loss to encourage the image features to be closely aligned with their corresponding ground truth labels. As for the sketch features, we

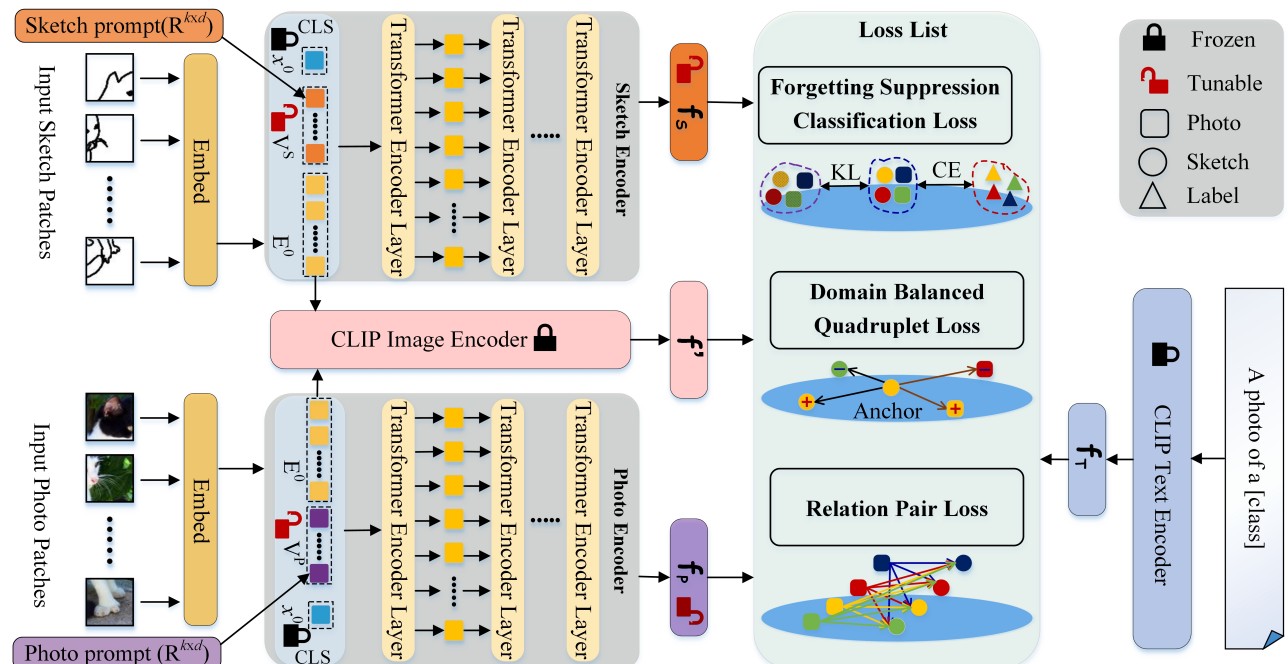

**Figure 3: The workflow of our method. Dr. CLIP employs a multi-branch network to handle sketches and photos separately, both initialized with the image encoder (ViT-B/32) of CLIP.**

consider their probability distribution:

$$P(y_i|s_i) = \frac{\exp\left(sim\left(f_{S_i}, f_{t_i}\right)/t\right)}{\sum_{j=1}^{N^T} \exp\left(sim\left(f_{S_i}, f_{t_j}\right)/t\right)} \quad (1)$$

Where $t$ represents temperature, and $sim(\cdot)$ denotes cosine similarity. The classification loss for sketches :

$$L_{\text{cls}}^S = \frac{1}{N^S} \sum_{i=1}^{N^S} -\log\left(P\left(y_i|s_i\right)\right) \quad (2)$$

The probability distribution of photo features is $P(y_i|p_i)$. The classification loss of the photos:

$$L_{\text{cls}}^P = \frac{1}{N^P} \sum_{i=1}^{N^P} -\log\left(P\left(y_i|p_i\right)\right) \quad (3)$$

To effectively suppress catastrophic forgetting, we aim to maintain relative consistency between the distribution of newly learned class features and the distribution of class features from existing knowledge [9]. The Encoder's predicted probability distribution for different classes as $d_i = \log\left(softmax\left(sim\left(f_i, f_t\right)\right)\right)$, the predicted probability distributions for sketches and photos as $d_s$ and $d_p$, and the distribution from existing knowledge as $d_s'$ and $d_p'$. Minimizing the KL divergence between the relative distance distributions of sketches for different classes:

$$L_{FS} = \frac{1}{N^S} \sum_{i=1}^{N^S} KL(d_s, d_s') + \frac{1}{N^P} \sum_{i=1}^{N^P} KL(d_p, d_p') \quad (4)$$

The sum of the aforementioned losses is referred to as the forgetting suppression classification loss ($L_{FSC}$).

$$L_{FSC} = L_{cls}^S + L_{cls}^P + \lambda_1 L_{FS} \quad (5)$$

$\lambda_1$ is a weight hyperparameter. $L_{FSC}$ can assist the model in learning category semantic knowledge, guiding the model to correctly recognize the categories of images.

## 3.4 Domain Balanced Quadruplet Loss

Triplet loss is commonly used to measure the similarity between positive and negative samples. A triplet typically consists of an anchor, a positive sample (a different instance of the same category as the anchor), and a negative sample (an instance from a different category than the anchor). The anchor is a sketch, and positive and negative samples are selected from the photo domain, denoted as $(f_s, f_p^+, f_p^-)$. The triplet loss is represented as follows:

$$L_{Tri}^P = \frac{1}{N^S} \sum_{i=1}^{N^S} \max\left\{sim\left(f_s, f_p^+\right) - sim\left(f_s, f_p^-\right) + m, 0\right\} \quad (6)$$

Where m=0.3 is a margin parameter, and $N^S$ represents the number of triplets. Taking domain-balanced sampling into account, the quadruplet loss extends the triplet framework by adding negative sketch samples (sketch instances from different categories than the anchor), creating new quadruplets $(f_s, f_p^+, f_p^-, f_s^-)$. The new item as follows:

$$L_{Tri}^S = \frac{1}{N^S} \sum_{i=1}^{N^S} \max\left\{sim\left(f_s, f_p^+\right) - sim\left(f_s, f_s^-\right) + m, 0\right\} \quad (7)$$

The new domain balanced quadruplet loss ($L_{DBQ}$) is represented as follows:

$$L_{DBQ} = L_{Tri}^P + L_{Tri}^S \tag{8}$$

$L_{DBQ}$ can guide the model in generating embedding representations where samples within the same category have smaller distances, while samples from different categories have larger distances. Such embedding representations facilitate accurate similarity measurement during the testing phase.

## 3.5 Relation Pair Loss

To facilitate the learning of semantic relationships between sketches and photos, it is essential to obtain sketch visual features that carry semantic associations. Following the domain-balanced approach, a set of image feature lists can be constructed.

$$Logits_S = \left\{ \left[ f_{s_i} \cdot f_i^T, \cdots, f_{s_i}^- \cdot f_i^T, \cdots \right], i = 1, \cdots, N^S \right\} \tag{9}$$

The photo visual features are represented as $Logits_P$. We score the similarity between each pair of features, resulting in a relationship matrix capturing the relationships between image pairs.

$$Score = sigmoid\left(sim\left(Logits_S, Logits_P\right)\right) \tag{10}$$

Following the rules for constructing the image feature lists, we can easily obtain the corresponding label lists. Assigning a value of 1 to indicate the same category and 0 for different categories results in the relation labels ($RL$). The relation pair loss ($L_{RP}$) can be represented as:

$$L_{RP} = \frac{1}{N} \sum_{i=1}^{N} \left(Score_i, RL_i\right)^2 \tag{11}$$

Where $N$ represents the number of images in the list.

$L_{RP}$ can guide the model in learning the similarity relationships between pairs of samples, aiding the model in capturing the relevance and ranking relationships between images more effectively.

## 3.6 Overall Objective

The complete loss function for the proposed method can be expressed as follows:

$$L_{total} = L_{FSC} + \lambda_2 L_{DBQ} + \lambda_3 L_{RP} \tag{12}$$

Where $\lambda_2$ and $\lambda_3$ are the weight hyperparameters. The optimization objective of the model is to find $(\theta^S, \theta^P, V^S, V^P)$ that satisfy:

$$\left(\theta^S, \theta^P, V^S, V^P\right) = \underset{\theta^s, \theta^p, V^s, V^p}{\arg min} \ L_{total} \tag{13}$$

## 4 EXPERIMENTS

### 4.1 Implementation Details

**Initialization.** The experiments are performed using PyTorch on a single 16GB Tesla V100 GPU. For the four zero-shot tasks, the input images are resized to 224*224. The encoder learning rate is 1e-4, prompt learning rate is 1e-3, batch size is 64, and prompts number is 3. To ensure that the initial values of each loss component are around 1.00, the weight hyperparameters $\lambda_1$, $\lambda_2$, and $\lambda_3$ are set to15 , 4, and 4, respectively. In all of the result tables, the red values indicate the best performance, while the blue values represent the second-best performance.

**Dataset.** Table 1 presents detailed quantity information and partition rules for the five datasets. All the comparative methods follow the same data partitioning protocol. For the ZS-CGSBIR task. **TU-Berlin Ext** [37]: It consists of 20,000 sketches from the TU-Berlin [7] dataset and an additional 204,489 real photos. This dataset exhibits significant class imbalance. **Sketchy Ext** [18]: To facilitate zero-shot tasks, 21 categories that do not overlap with the ImageNet dataset were designated as the test set [34], while the remaining 104 categories were used for training. **QuickDraw** [12]: The sketches are created by volunteers from around the world within a specified time limit. For the ZS-FGSBIR task. **QMUL-ChairV2** [26] and **QMUL-ShoeV2** [35] are two commonly used fine-grained sketch datasets. For the ZS-CDSBIR task, the same dataset settings as [17] were employed. **S**, **T**, and **Q** represent the Sketchy Ext, TU-Berlin Ext, and QuickDraw datasets, respectively. **S→T(21)** indicates training on the Sketchy Ext training set and testing on the TU-Berlin Ext dataset, selecting 21 categories from TU-Berlin Ext that do not appear in the Sketchy Ext training set. There are no overlapping categories between the test and training sets, fulfilling the requirements of zero-shot learning. For the GZS-SBIR task, the dataset processing method from work [13] was adopted. The images of 20% of the seen categories in the training set were augmented into the test set, creating generalized retrieval datasets ( **TU-Berlin Ext-G**, **Sketchy Ext-G**) containing both seen and unseen categories.

**Evaluation Metrics.** To ensure fairness, we adopt three evaluation metrics that are consistent with previous works [13, 17, 25]. Retrieval Precision on Top K (e.g., **P@100**, **P@200**) measures the proportion of correct results that match the query sketch among the top k retrieval results. Retrieval Accuracy (e.g., **acc.@1**, **acc.@10**) measures the accuracy of the best matching result for a given query sketch. Mean Average Precision (e.g., **mAP@100**, **mAP@all**) quantifies a comprehensive assessment of the average accuracy across different numbers of retrieval results.

### 4.2 Experimental Analysis of ZS-CGSBIR Task

**Competitors.** The select competitors are ZS-TVT, ZS-PSKD[ViT], ZS-Sketch3T, TCN, STL, ACNet, ZSE-Ret, and CLIP-all, which are the ZS-CGSBIR SOTA methods. These methods are based on zero-shot settings. The experimental results for CLIP-all are from the public code, while others are from original papers.

It is evident from Table 2 that our proposed method achieves the highest results across all metrics and three datasets. For example, our method outperforms the second-best value with a **3.1%** improvement in P@100 on the TU-Berlin Ext dataset, a **2.7%** increase in mAP@200 on the Sketchy Ext dataset, and a **4.8%** enhancement in mAP@all on the QuickDraw dataset.

We select the top-performing three methods to showcase real retrieval cases, as illustrated in Figure. 4. It is visually evident that our proposed method exhibits fewer erroneous results compared to the other two methods. Furthermore, our retrieval results are visually more similar to the query sketches. For example, for the '*duck*' query sketch, our method prioritizes returning photos with consistent poses and actions, rather than solely focusing on photos with matching categories. For the '*scissors*' query sketch, our method mistakenly returns an photo of a sword. However, upon closer observation, it is apparent that this sword photo bears a high

Table 1: Datasets used in four ZS-SBIR tasks: quantity, partition information, * denotes computed mean values, not actual values.

| Items | ZS-CGSBIR | | | ZS-FGSBIR | | ZS-CDSBIR | | | | GZS-SBIR | |
| --- | --- | --- | --- | --- | --- | --- | --- | --- | --- | --- | --- |
| | TU-Berlin Ext | Sketchy Ext | QuickDraw | ChairV2 | ShoeV2 | S→T(21) | S→Q(11) | T→S(8) | T→Q(10) | TU-Berlin-G | Sketchy-G |
| Classes | 250 | 125 | 110 | 400 | 2,000 | 125 | 115 | 228 | 230 | 250 | 125 |
| Training classes | 220 | 104 | 80 | 300 | 1,800 | 104 | 104 | 220 | 220 | 220 | 104 |
| Testing classes | 30 | 21 | 30 | 100 | 200 | 21 | 11 | 8 | 10 | 74 | 42 |
| Sketches | 20,000 | 75,471 | 330,000 | 1,275 | 6,730 | 64,467 | 95,798 | 22,127 | 47,600 | 20,000 | 75,471 |
| Sketches per class | 80 | 600* | 3,000 | 2~4 | 1~4 | 603*/80 | 603*/3,000 | 80/566* | 80/3,000 | 80 | 600* |
| Photos | 204,489 | 73,002 | 204,000 | 400 | 2,000 | 79,084 | 82,106 | 182,156 | 197,380 | 204,489 | 73,002 |
| Photos per class | 818* | 584* | 1,854* | 1 | 1 | 600*/787 | 600*/1,778* | 811*/474* | 811*/1,902* | 818* | 584* |

Table 2: Comparison of the proposed method and competitors on the TU-Berlin Ext, Sketchy Ext and QuickDraw datasets in the ZS-CGSBIR task.

| Competitors | TU-Berlin Ext | | Sketchy Ext | | QuickDraw | |
| --- | --- | --- | --- | --- | --- | --- |
| | mAP@all | P@100 | mAP@200 | P@200 | mAP@all | P@200 |
| ZS-TVT(AAAI2022)[29] | 0.484 | 0.662 | 0.531 | 0.618 | 0.149 | 0.293 |
| ZS-PSKD[ViT](ACM MM2022)[31] | 0.502 | 0.662 | 0.560 | 0.645 | 0.150 | 0.298 |
| ZS-Sketch3T(CVPR2022)[28] | 0.507 | 0.671 | 0.579 | 0.648 | - | - |
| TCN(TPAMI2022)[30] | 0.495 | 0.616 | 0.516 | 0.608 | 0.140 | 0.231 |
| STL(AAAI2023)[10] | 0.402 | 0.498 | 0.634 | 0.538 | - | - |
| ACNet(TCSVT2023)[24] | 0.577 | 0.658 | 0.517 | 0.608 | - | - |
| ZSE-Ret(CVPR2023)[17] | 0.569 | 0.637 | 0.504 | 0.602 | 0.142 | 0.202 |
| CLIP-all(CVPR2023)[25] | 0.656 | 0.732 | 0.713 | 0.680 | 0.194 | 0.225 |
| Our | 0.685 | 0.763 | 0.74 | 0.706 | 0.242 | 0.312 |

Table 3: Comparison of the proposed method and competitors on the ChairV2 and ShoeV2 datasets in the ZS-FGSBIR task.

| Task | Competitors | ChairV2 | | ShoeV2 | |
| --- | --- | --- | --- | --- | --- |
| | | acc.@1 | acc.@10 | acc.@1 | acc.@10 |
| FGSBIR | CC-DG(CVPR2019)[21] | 0.5421 | 0.8823 | 0.338 | 0.7786 |
| | Triplet-RL(2020 CVPR)[3] | 0.5654 | 0.8961 | 0.341 | 0.7882 |
| | MPA(2021CVPR)[2] | 0.533 | 0.875 | 0.334 | 0.807 |
| ZS-FGSBIR | ZSE-Ret(CVPR2023)[17] | 0.6431 | 0.926 | 0.3171 | 0.6733 |
| | ZSE-RN(CVPR2023)[17] | 0.6334 | 0.9453 | 0.2206 | 0.6977 |
| | CLIP-all(CVPR2023)[25] | 0.3183 | 0.8006 | 0.1216 | 0.4895 |
| | Our | 0.5177 | 0.9325 | 0.255 | 0.740 |

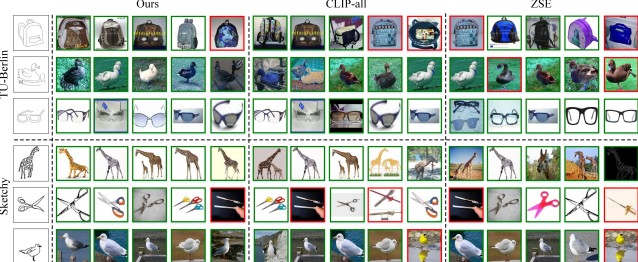

Figure 4: Comparative analysis of query sketches and top 5 results on two coarse-grained datasets. Correct results are marked in green, while incorrect results are marked in red.

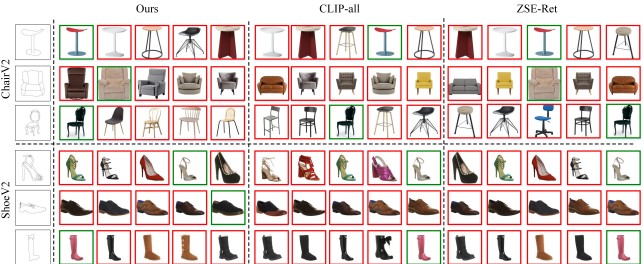

Figure 5: Comparative analysis of query sketches and top 5 results on two fine-grained datasets. Correct results are marked in green, while incorrect results are marked in red.

resemblance to the local features of the query sketch. In contrast, the erroneous results returned by other methods do not exhibit such visual similarity.

## 4.3 Experimental Analysis of ZS-FGSBIR Task

**Competitors.** We chose CC-Gen, Triplet-RL, and MPA as competitors, which are the FGSBIR methods. And consider two variants of ZSE (ZSE-Ret, ZSE-Rn) and CLIP-all as competitors, which are the ZS-FGSBIR SOTA methods. ZSE and CLIP-all results are from public code, while others are from original papers.

The proposed method achieves ideal results across all metrics and two fine-grained datasets. Compared to the ZS-FGSBIR method, our approach also performs admirably. For example, on the ChairV2 dataset, the difference in acc.@10 compared to the best mAP is only **1.28%**, but compared to the baseline model CLIP-all, acc.@1 and acc.@10 have improved by **19.94%** and **13.19%**, respectively. On the ShoeV2 dataset, acc.@10 is **4.23%** higher than the second-best mAP. Although our method is not optimal, it still achieves comparable performance to the best methods. We select the top three performing methods and showcase real retrieval cases in

Figure. 5. It is evident that our method exhibits higher accuracy in TOP-1 and TOP-5 results compared to the other two methods. Additionally, in the TOP-5 results, our method consistently ranks correct results higher. For example, in the ChairV2 dataset, with a query sketch of "leg makeup chair," our method retrieves the correct result in the first position, while the other two methods are ranked third and fifth, respectively.

## 4.4 Experimental Analysis of ZS-CDSBIR Task

**Competitors.** We select DSN, ZSE-Rn, and CLIP-all, which are the ZS-CDSBIR SOTA methods. CLIP-all results are from public code, while others are from original papers.

It is evident that the proposed method achieves the highest results across all metrics and the three datasets. As shown in Table 4, our method exhibits an average improvement of **9.02%** and **8.85%** in P@100 and mAP@all, respectively, across the four datasets. The most significant improvement is observed in the S→T(21) dataset, where our method outperforms the second-best method by **15.38%** in P@100 and **19.14%** in mAP@all, achieving highly favorable retrieval performance. On the T→Q(10) dataset, our method achieves

Table 4: Comparison of the proposed method and competitors on the newly generated datasets in the ZS-CDSBIR task.

| Competitors | S→T(21) | | S→Q(11) | | T→S(8) | | T→Q(10) | |
|---|---|---|---|---|---|---|---|---|
| | P@100 | mAP@all | P@100 | mAP@all | P@100 | mAP@all | P@100 | mAP@all |
| CC-DG(CVPR2019)[21] | 0.434 | 0.308 | 0.227 | 0.156 | 0.693 | 0.624 | 0.296 | 0.231 |
| DSN(IJCAI2021)[32] | 0.469 | 0.356 | 0.178 | 0.149 | 0.654 | 0.613 | 0.246 | 0.218 |
| ZSE-RN(CVPR2023)[17] | 0.59 | 0.476 | 0.228 | 0.338 | 0.816 | 0.746 | 0.376 | 0.273 |
| CLIP-all(CVPR2023)[25] | 0.5519 | 0.4799 | 0.3595 | 0.3584 | 0.8573 | 0.8306 | 0.5192 | 0.4621 |
| Our | 0.7438 | 0.6713 | 0.5149 | 0.4733 | 0.8901 | 0.8588 | 0.5380 | 0.4816 |

Table 5: Comparison of the proposed method and competitors on the Sketchy-G, and TU-Berlin-G datasets in the GZS-SBIR task.

| Competitors | Sketchy-G | | | | TU-Berlin-G | | | |
|---|---|---|---|---|---|---|---|---|
| | P@100 | P@200 | mAP@200 | mAP@all | P@100 | P@200 | mAP@200 | mAP@all |
| SEM-PCYC(CVPR2019)[5] | 0.3405 | 0.3229 | 0.3749 | 0.2355 | 0.2192 | 0.2098 | 0.2411 | 0.1415 |
| Doodle2Search(CVPR2020)[4] | 0.3747 | 0.3348 | 0.4214 | 0.3104 | 0.0491 | 0.0448 | 0.0685 | 0.0439 |
| OCEAN(ICME2020)[40] | 0.548 | 0.443 | 0.547 | 0.445 | 0.341 | 0.319 | 0.369 | 0.312 |
| STL(AAAI2023)[10] | 0.584 | 0.541 | 0.636 | 0.533 | 0.503 | 0.472 | 0.532 | 0.406 |
| AMF(TIP2022)[13] | 0.4383 | 0.4249 | 0.4545 | 0.38 | 0.2376 | 0.2255 | 0.2587 | 0.1622 |
| CLIP-all(CVPR2023)[25] | 0.6986 | 0.6862 | 0.6593 | 0.6593 | 0.6846 | 0.6545 | 0.7063 | 0.6077 |
| Our | 0.7288 | 0.7135 | 0.739 | 0.6836 | 0.7219 | 0.6894 | 0.7432 | 0.6529 |

Table 6: Different variants in the ZS-CGSBIR task are evaluated based on four metrics on the category-level dataset.

| Competitor | TU-Berlin Ext | | | | Sketchy Ext | | | | QuickDraw | | | |
|---|---|---|---|---|---|---|---|---|---|---|---|---|
| | P@100 | P@200 | mAP@200 | mAP@all | P@100 | P@200 | mAP@200 | mAP@all | P@100 | P@200 | mAP@200 | mAP@all |
| baseline | 0.732 | 0.697 | 0.7564 | 0.656 | 0.704 | 0.6802 | 0.7131 | 0.6427 | 0.225 | 0.2254 | 0.1913 | 0.1943 |
| $w/o$ FSC | 0.7598 | 0.7129 | 0.7744 | 0.6798 | 0.7282 | 0.7025 | 0.7384 | 0.6697 | 0.2892 | 0.2858 | 0.3104 | 0.2312 |
| $w/o$ DBQ | 0.7512 | 0.7073 | 0.7654 | 0.6766 | 0.7175 | 0.6932 | 0.7283 | 0.6633 | 0.2721 | 0.2675 | 0.2946 | 0.2119 |
| $w/o$ RP | 0.7403 | 0.6934 | 0.7557 | 0.6542 | 0.7095 | 0.6825 | 0.7197 | 0.6467 | 0.2886 | 0.2855 | 0.3103 | 0.2228 |
| Our | 0.7634 | 0.7174 | 0.7756 | 0.6847 | 0.7311 | 0.7064 | 0.7395 | 0.6761 | 0.2911 | 0.286 | 0.3125 | 0.2424 |

a relative improvement of **1.88%** in P@100 and **1.95%** in mAP@all compared to the second-best method. Furthermore, we discover an interesting phenomenon. On the QuickDraw dataset, models trained on the T→Q(10) and S→Q(11) datasets surprisingly exhibit better retrieval performance than models trained solely on QuickDraw itself. Upon analysis, we find that the sketch quality in QuickDraw is significantly lower compared to the other two datasets. QuickDraw provides a larger number of sketch samples, but it also introduces a considerable amount of noise.

## 4.5 Experimental Analysis of GZS-SBIR Task

**Competitors.** We select SEM-PCYC, Doodle2Search, OCEAN, STL, AMF, and CLIP-all as competitors, which are the GZS-SBIR SOTA methods. CLIP-all results are from public code, while others are from original papers.

As shown in Table 5, it is obvious to observe that the proposed method achieves the highest result on all metrics and two datasets. On the two datasets, our method achieves an average improvement of **3.38%**, **3.11%**, **5.83%**, and **3.48%** in P@100, P@200, mAP@200, and mAP@all, respectively, compared to the second-best value. Compared to the AMF method, our method achieves highly favorable retrieval performance. Surprisingly, combining the results in Table 2 and Table 5, our method does not significantly decrease the retrieval accuracy when additional seen classes are introduced in the test data. This indicates that our method possesses stronger

Table 7: Different variants in the ZS-FGSBIR task are evaluated based on four metrics on the fine-grained datasets.

| Competitor | ChairV2 | | | ShoeV2 | | |
|---|---|---|---|---|---|---|
| | acc@1 | acc@5 | acc@10 | acc@1 | acc@5 | acc@10 |
| baseline | 0.3183 | 0.6656 | 0.8006 | 0.1216 | 0.3423 | 0.4895 |
| $w/o$ FSC | 0.4598 | 0.8135 | 0.9196 | 0.2252 | 0.5375 | 0.6982 |
| $w/o$ DBQ | 0.3923 | 0.7781 | 0.9164 | 0.1967 | 0.515 | 0.6802 |
| $w/o$ RP | 0.434 | 0.8006 | 0.9035 | 0.2267 | 0.5225 | 0.6907 |
| Our | 0.5177 | 0.8457 | 0.9325 | 0.255 | 0.565 | 0.74 |

learning and adaptation capabilities, enabling it to handle the similarities and differences between seen and unseen classes and address the challenges of complex task scenarios.

## 4.6 Ablation Study

The symbol '$w/o$' indicates the removal of a specific loss term. We evaluate the importance of each loss term. Table 6 and Table 7 provide a statistical comparison of these variants across five different metrics, both on the category-level and the fine-grained dataset.

According to the results in Table 6, our method achieve an average improvement of **2.49%**, **2.83%**, and **7.40%** over the baseline method on the TU-Berlin Ext, Sketchy Ext, and QuickDraw datasets, respectively. This indicates the effectiveness of our optimization strategies, specifically highlighting the contribution of each loss

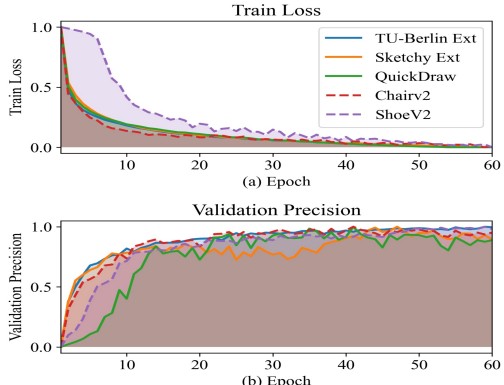

Figure 6: The loss and precision curves of the proposed method during training on five datasets

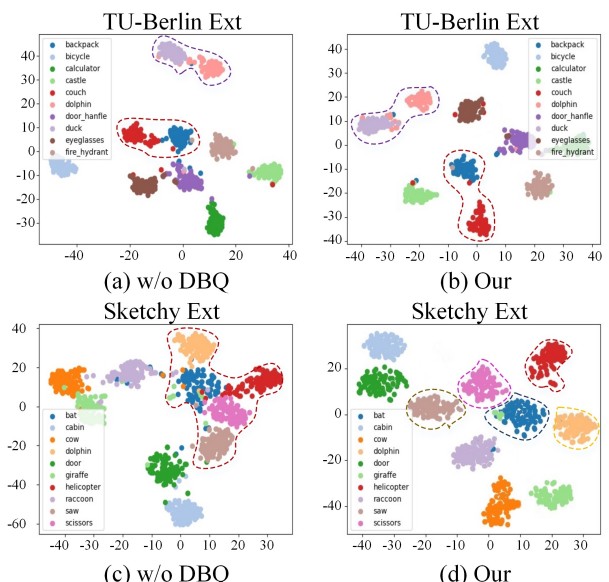

Figure 7: The class feature distributions (t-SNE) generated by the $w/o$ DBQ and our method on the TU-Berlin Ext and Sketchy Ext validation sets.

term in enhancing category-level sketch retrieval performance. Analyzing the average improvement across different datasets, we observed that the impact on the final retrieval accuracy followed the order of RP>DBQ>FSC. This suggests that improving the matching between image-text pairs is particularly beneficial for enhancing the semantic understanding ability of the model on coarse-grained data. It is worth noting that the design objective of the FSC loss is to prevent forgetting knowledge during the training process and mitigate overfitting. As shown in Figure 6, we visualize the curve for verifying accuracy during the training process again.

As shown in Table 7, our method achieve an average improvement of **4.99%** and **4.45%** over the baseline on the ChairV2 and ShoeV2 datasets, respectively, demonstrating the effectiveness of

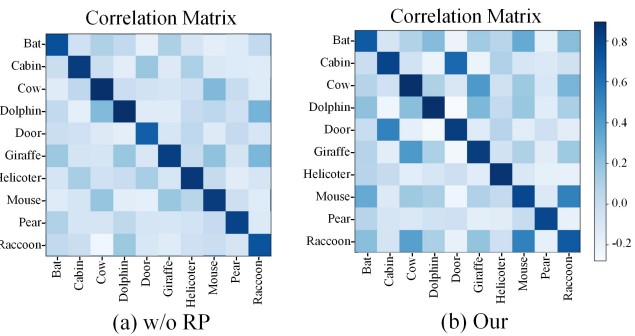

Figure 8: Visualization of the feature correlation matrix for sketch-photo pairs of ten classes on the Sketchy Ext.

our approach. Specifically, each loss term contributed to better fine-grained sketch retrieval performance. Analyzing the average improvement across different datasets, we observed that the impact on the final retrieval accuracy followed the order of DBQ>RP>FSC. This indicates that enhancing the matching between sketch-photo pairs is particularly beneficial for improving the discriminative ability of the model on fine-grained data. Additionally, we attempted to illustrate the impact of each loss term on the model through visualizations. Figure. 6 shows that the FSC loss eliminates the overfitting phenomenon of the CLIP model during training on the ChairV2 and ShoeV2 datasets. In Figure. 7, we visualize the feature projections of two validation sets using the $w/o$ DBQ model and our model (with 100 randomly select return samples per class for Sketchy). Our model successfully differentiated the overlapping class data that the $w/o$ DBQ model could not. In Figure. 8, we select the mean features of sketches and photos for 10 classes from the validation set and plot the correlation matrix between sketch-photo pairs. The $w/o$ RP model achieve a higher correlation for intra-class pairs compared to inter-class pairs. Therefore, our method not only achieve a higher correlation for intra-class pairs but also capture similarities between similar pairs, such as 'door' and 'cabin', 'mouse' and 'raccoon', which have inherent visual similarities. This strategy helps align the ordering of the returned results more closely with the principles of human visual sorting.

## 5 CONCLUSION AND FUTURE WORK

Most existing ZS-SBIR models are primarily tailored for specific tasks. This limitation poses a significant barrier to their widespread implementation in real-world environments. The CLIP-driven universal framework (Dr. CLIP) is an attempt to lift zero-shot sketch-based image retrieval into the era of universal models. Our principal contributions are a new task (all variants of ZS-SBIR with just one network) and model (Dr. CLIP) that make this leap possible. The model is evaluated on five datasets across four experimental environments, and comparisons are made with recent state-of-the-art methods. The experimental results unequivocally demonstrate the versatility and superiority of our approach. In future research, we will further explore and exploit the synergies between visual language models and retrieval tasks to delve deeper into this field.

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
