# OpenReview forum: "Dr. CLIP: CLIP-Driven Universal Framework for Zero-Shot Sketch Image Retrieval"
_acmmm.org/ACMMM/2024/Conference — MM2024 Poster_

### Official Review · Reviewer_X2wq · 2024-05-13

**Rating:** 4
**Confidence:** 4

**Summary:**

Inspired by the impressive generalization ability of the Contrastive Language-Image Pretraining (CLIP) model, this paper proposes a CLIP-driven universal framework (Dr.CLIP), which leverages prompt learning to guide the synergy between CLIP and Zero-Shot Sketch-Based Image Retrieval (ZS-SBIR). This paper decomposes the synergy into classification learning, metric learning, and ranking learning, as well as respectively introduces three kinds of loss including Forgetting Suppression Classification Loss, Domain Balanced Quadruplet Loss, and Relation Pair Loss to enhance learning effectiveness. This is the first universal framework that covers four different ZS-SBIR subtasks. The Dr.CLIP framework has demonstrated superior performance and achieved state-of-the-art results on three coarse-grained datasets and two fine-grained datasets.

**Strengths:**

1. Overall: The proposed Dr.CLIP universal framework utilizes the generalization ability of Contrastive Language-Image Pretraining (CLIP) and loss design to produce a more general retrieval model capable of managing various specialized subtasks in the Zero-Shot Sketch-Based Image Retrieval (ZS-SBIR) task. The effectiveness of the Dr.CLIP method is evaluated by comparing it against several state-of-the-art baselines on three coarse-grained datasets and two fine-grained datasets.
2. Experiment:
1) Dataset: The dataset conditions and partitioning details are clearly outlined.
2) Experiments: The experiment in this paper is sufficient and effectively demonstrates the superiority of the proposed method.
3. Writing: The proposed method is explained in a straightforward manner, so this paper is easy to understand and easy to follow.

**Limitations:**

1. Writing error. Line 306 “The test set comprises sketches and photos of the unseen classes, denoted as 𝐷𝑡𝑒𝑠𝑡 = {𝑆𝑠𝑒𝑒𝑛, 𝑃𝑠𝑒𝑒𝑛}.” There is a writing error concerning the dataset references; it should be 𝐷𝑡𝑒𝑠𝑡 = {𝑆un𝑠𝑒𝑒𝑛, 𝑃un𝑠𝑒𝑒𝑛 }
2. Experimental results analysis is insufficient. The proposed method does not achieve the best or even second-best performance on the ChairV2 in the ZS-FGSBIR task. Is there any reason or analysis for the gap between the proposed method and the state-of-the-art methods?
3. Limited multimodal processing and integration. Although the paper utilizes the fusion of visual and textual modalities, the treatment of the textual modality is limited to converting “category + prompt template” into text embeddings using CLIP’s frozen text encoder. These embeddings are then integrated into the loss design. From my limited perspective, this approach does not significantly enhance the processing and integration of multimodal data.

**Suitability:**

2

---

### Official Review · Reviewer_4bz5 · 2024-05-24

**Rating:** 4
**Confidence:** 3

**Summary:**

This work makes a contribution to the field of multimodal processing by embracing the paradigm shift in Zero-Shot Sketch-Based Image Retrieval (ZS-SBIR) and developing a more general retrieval model capable of managing various specialized scenarios

**Strengths:**

1. This paper proposes a CLIP-driven universal framework named Dr. CLIP, which is designed to handle multiple zero-shot sketch-based image retrieval tasks. This is the first universal framework to cover four different subtasks of zero-shot sketch-based image retrieval.
2. The forgetting suppression idea can prevent catastrophic forgetting and constrain the feature distribution of new classes and a domain balanced loss is proposed to address sample imbalance and establish effective cross-domain correlations.
3. Four experimental scenarios for zero-shot sketch-based image retrieval on three coarse-grained datasets and two fine-grained datasets demonstrate that the proposed method achieves the best retrieval accuracy in each task, showcasing its effectiveness and versatility.
4. Visualization and ablation experiments are conducted to analyze the contribution of each loss term, demonstrating the motivation and effectiveness of the proposed method.
Weakness

**Limitations:**

1.	Regarding the necessity of the zero-shot cross-dataset sketch-based image retrieval and the Generalized zero-shot sketch-based image retrieval (GZS-SBIR), I believe that these two scenarios are essentially covered by other scenarios, and there is no need for specialized research on them.
2.	The experimental section lacks analysis of hyperparameters, thus the contribution analysis of different modules to the model is insufficient.
3.	The loss functions proposed in the experiment are mostly variants of conventional losses, such as the Domain Balanced Quadruplet Loss, which is a conventional contrastive learning loss. In terms of innovation, it may be slightly lacking.

**Suitability:**

2

---

### Official Review · Reviewer_vF75 · 2024-05-31

**Rating:** 2
**Confidence:** 4

**Summary:**

The paper focuses on Zero-Shot Sketch-Based Image Retrieval (ZS-SBIR) task. Specifically, the authors use the prompt tuning method and CLIP to improve the performance. Experiments prove the effectiveness of the proposed method.

**Strengths:**

1. The paper is easy to follow. Most of the paper is well-written and very clear.

2. Experiments are exhaustive and prove the effectiveness of the proposed method.

**Limitations:**

1. My biggest concern is the lack of novelty. I am pretty familiar with the research topics including in the paper, such as CLIP, prompt tuning, image retrieval, zero-shot, and metric learning loss. As far as I am concerned, it contains NO novelty at all. Specifically, (1) the CLIP for the ZS-SBIR task is well-studied in [1], although your performance maybe a little better; (2) the prompt tuning technique has been well-explored in the context of computer vision, and you do not propose any new insight to this; (3) the used three losses, such as domain-balance loss, are common to me, and it is straightforward to use them in your task.

2. ZS-SBIR task is not proposed by you, and thus it is unnecessary to categorize and discuss it in Fig. 1. Fig. 1 should be a teaser image to introduce the core of your paper to the readers. Although your writing is good, this may come from the polishment from ChatGPT. It would help if you focused more on the academic content rather than the grammatical and English usage. In addition, what do you mean in Abstract by: https://github.com/xxxxxx.git?

In conclusion, I appreciate the hardworking authors make into this article, but it should be rejected by a top-tier conference.


[1] CLIP for All Things Zero-Shot Sketch-Based Image Retrieval, Fine-Grained or Not

**Suitability:**

2

---

### Official Review · Reviewer_BrEX · 2024-06-01

**Rating:** 3
**Confidence:** 3

**Summary:**

This paper presents a CLIP based general ZS-SBIR model aiming to manage various specialized ZS tasks. It is a multi-branch network model based on CLIP image encoder and text encoder and three key components: forgetting suppression, domain balanced loss and pair-relation strategy are introduced to enhance learning effectiveness. Experiments are conducted on 5 datasets on 4 ZS-SBIR tasks. The experimental results are impressive.

**Strengths:**

The idea is interesting to explore the general model for various tasks of ZS-SBIR. And the experimental results are impressive compared with the previous works as the counterparts in the experiments.

The three key components sound reasonable for dealing with the general model for ZS-SBIR.

**Limitations:**

The problem of general ZS-SBIR is discussed on some previous works such as ZSE and CLIP-all，especially CLIP-all is very relevant to this work. Although this work extends the tasks from 2 to 4, but the novelty of the paper seems not very strong.

The writing of the paper is not very well. The motivation of the paper is not very clear in the introduction part. And why choose the three methods to solve the problem needs more discussion in the introduction. In my opinion, the explanations of the four ZS-SBIR tasks seem redundant somehow in introduction, and the motivations of the work are missing / insufficient.

The working flow of the model given in Figure 3 is not clear. The prompt is important for CLIP like model, but this part of the model in Section 3.2 is too simple to understand how it works in this paper. And the issues of model training given in Section 4.1 also need more details.

**Suitability:**

2

---

### Official Review · Reviewer_o7Eb · 2024-06-09

**Rating:** 5
**Confidence:** 3

**Summary:**

Dr. CLIP provides an innovative framework for Zero-Shot Sketch Image Retrieval by introducing Forgetting Suppression Classification Loss, Domain Balanced Quadruplet Loss, and Relation Pair Loss, which enable effective fine-grained feature capture and cross-domain alignment. Dr. CLIP has achieved state-of-the-art performance on multiple datasets and tasks, demonstrating significant performance improvements.

**Strengths:**

1. In this paper, the authors propose a CLIP-driven universal framework for multiple zero-short sketch-image retrieval tasks while exploring the synergy between CLIP and ZSSBIR. A multimodal feature representation is learned to capture instance relationships.
2. The logic is clear and easy to follow.
3. Extensive experimental results show the effectiveness of the proposed model with four ZS-SBIR experimental scenarios on three coarse-grained datasets.

**Limitations:**

1. The explanation of the innovations is not detailed enough, especially regarding the design principles and expected improvements of each innovation. Specifically, for the Forgetting Suppression Classification Loss, it is recommended to further explain its mechanism for preventing forgetting and how it compares to existing methods for preventing forgetting.
2. The description of the experimental setup and hyperparameter selection is not detailed enough, which may affect the reproducibility of the experiments. It is suggested to provide detailed hyperparameter selection and training details to ensure that other researchers can replicate these experiments and verify the results.
3. The model's performance in ZS-FGSBIR tasks is suboptimal, possibly due to its inadequacy in capturing subtle differences between different subcategories. It is recommended that the authors conduct an in-depth analysis of the model's shortcomings in fine-grained feature capture and add a detailed discussion of this issue in the paper, explaining the possible reasons and the limitations of the current model.

**Suitability:**

3

---

### Meta-Review · Area_Chair_mT5W · 2024-07-07

**Recommendation:** Accept (Poster)
**Confidence:** 3

**Metareview:**

This paper proposes a CLIP-based matching framework that learns multimodal feature representation for zero-shot sketch-image retrieval (ZS-SBIR). In total, there are 6 reviewers with mixed ratings (4 positive and 2 negative). The main concerns from the reviewers with negative rating are mainly about the novelty and the writing quality. Particularly, Reviewer vF75 pointed out that CLIP-ALL[1] already explored CLIP for ZS-SBIR, prior to this paper. In response, the authors clarified that they actually adopt CLIP-ALL as the baseline and achieve remarkable improvement on 4 benchmarks. Given the detailed comparison (between the proposed method and CLIP-ALL) and the non-trivial improvement, I am convinced of the novelty and effectiveness of this method. However, it is indeed inappropriate for the authors call their method as "CLIP-Driven Universal Framework", since CLIP-ALL already explored CLIP for ZS-SBIR. I would like to suggest the authors revise the title and some statement regarding the contribution of this paper, so that this paper will be acceptable.